# Enhancing Wrist Fracture Detection through LLM-Powered Data Extraction and Knowledge-Based Ensemble Learning

**Serge Vasylechko**[1,2] [ID]                    SERGE.VASYLECHKO@CHILDRENS.HARVARD.EDU
[1] *Quantitative Intelligent Imaging Laboratory, Department of Radiology, Boston Children's Hospital, Longwood Avenue, Boston, MA 02115*
[2] *Harvard Medical School, Longwood Avenue, Boston, MA 02115*

**Andy Tsai**[1,2]                               ANDY.TSAI@CHILDRENS.HARVARD.EDU
**Onur Afacan**[1,2]                             ONUR.AFACAN@CHILDRENS.HARVARD.EDU
**Sila Kurugol**[1,2]                            SILA.KURUGOL@CHILDRENS.HARVARD.EDU

**Editors:** Under Review for MIDL 2025

## Abstract

The accuracy and generalization of deep learning models for fracture detection and classification in wrist radiographs is often limited by the scarcity of high-quality annotated data and class imbalances. Traditional annotation methods are time-consuming, expensive and prone to inter-observer variability (Rajpurkar et al., 2017). To address these challenges, we developed an automated, cost-free approach to extract structured information from radiology reports, such as fracture type, location and severity. Our technique incorporates methods introduced by MedPrompt (Nori et al., 2023), and leverages domain expertise for group based sampling (Khan et al., 2024). Using these structured language labels alongside a pre-trained YOLO v7 backbone (Nagy et al., 2022; Ciri, 2023), which initially demonstrated low accuracy scores on our clinical data, we were able to selectively finetune the model in pseudo-blind manner. This approach utilized the extracted language labels without requiring expert annotations for training. We curated a large dataset of almost 3,000 pediatric wrist X-ray images and their corresponding radiology reports. Validation and testing were conducted on a smaller subset of 300 expert-annotated images. Our findings indicate that this pseudo-blind training strategy significantly enhances the base accuracy of the pre-trained model, achieving performance comparable to models finetuned with meticulously labeled expert annotations. Specifically, we improved the mean Average Precision (mAP) detection score for true positives related to fractures from 76% to 83%. Additionally, we observed improvements in precision and recall metrics for fracture detection. By integrating prompt-based information extraction with knowledge-based grouping, we achieved a robust and effective model for fracture detection. Our code and supplementary material are provided at www.github.com/sergeicu/yolostructured.

**Keywords:** LLM, object detection, limited data, weak labels

## 1. Introduction

Fractures represent one of the most common injuries in children, accounting for approximately 25% of all pediatric injuries (Butler et al., 2023). Accurate and timely diagnosis is crucial, as delays can lead to significant complications including improper healing and long-term functional impairments (Tadepalli et al., 2022). Despite their frequency, pediatric fractures pose unique diagnostic challenges due to developing anatomy (Marsh et al., 2007),

with X-rays remaining the source of common diagnostic errors in emergency departments (Newman-Toker et al., 2023).

The diagnostic performance of deep learning methods in fracture detection remains constrained by insufficient expertly annotated datasets and class distribution skewness. Traditional annotation methods requiring bounding boxes are time-consuming and yield inter-observer variability, as radiologists typically only mark areas with arrows or describe findings in reports (Martín-Noguerol et al., 2024). This particularly affects rare fracture patterns and child-specific factors such as growth plates that may appear like fractures to detection algorithms (Marsh et al., 2007).

While data scarcity poses challenges, fine-tuning pre-trained models offers promising solutions. YOLO architectures demonstrate notable efficiency, requiring as few as 2,000 images for medical applications (Montalbo, 2020), with fine-tuning scenarios achieving optimal performance using only 200-300 samples (Li et al., 2018). Recent approaches have explored pseudo-labeling to reduce annotation burden, such as leveraging Segment Anything Model (SAM) (Mazurowski et al., 2023) for creating labels from weakly supervised regions (Keuth et al., 2024).

The emergence of vision language models (VLMs) and large language models (LLMs) has enhanced contextual understanding of both image and text modalities. Research has shown that VLM-based scene understanding can improve YOLO object detection performance (Rouhi et al., 2025), while methods integrating bounding box predictions with language models show promise (Zang et al., 2024).

This work explores how LLMs can extract structured information from radiology reports to support YOLO training through knowledge-based grouping. By combining prompt-based information extraction with traditional fracture detection, we develop a more robust model for identifying accurate pseudo-labels, enabling improved pediatric fracture detection without additional expert annotation costs.

## 2. Method

### 2.1. Structured Information Extraction System

A comprehensive natural language processing pipeline was developed to extract structured information from radiological reports. The core utilized the Anthropic API (Anthropic, 2024) given its advanced performance in medical domain tasks (Kurokawa et al., 2024).

#### 2.1.1. Schema Validation

The following steps were implemented to ensure that the responses from the LLM adhered to a strict predictable structured response pattern: 1. Report Pre-processing: The initial step involved removing non-standard characters and normalizing medical abbreviations according to RADLEX terminology standards (Langlotz et al., 2006). 2. Schema Design: A schema was created to capture the hierarchical relationships between anatomical structures, fracture characteristics, and associated findings. 3. Core Extraction: The main extraction process utilized explicitly defined JSON structures passed to the LLM system, with response formatting managed through prefilled templates. 4. Type Validation: Type validation was enforced using Pydantic (Colvin, 2017) to ensure structured response patterns.

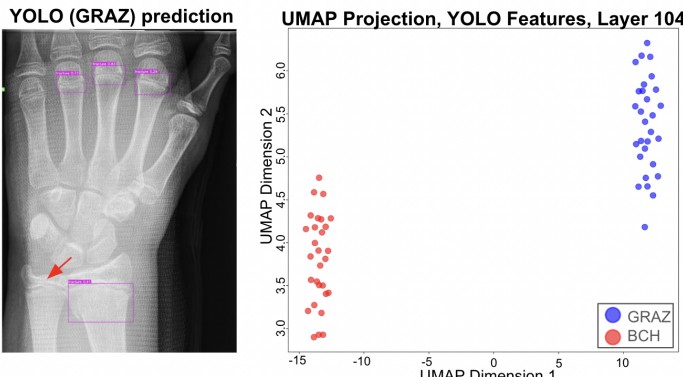

Figure 1: A. YOLO pre-trained on 20,000+ pediatric X-ray images from elsewhere correctly interpreted Salter-Harris II fracture of distal radius (pink, bottom) on our pediatric dataset (BCH), but it missed distal ulna fracture (red arrow) and incorrectly annotated three metacarpals as fractures (pink, top). B. UMAP projection of the YOLO features for 30 images from BCH and GRAZ datasets shows substantial domain shift.

2.1.2. PROMPT ENGINEERING AND SAFEGUARDS

To minimize hallucination risks, we implemented: (1) Chain-of-Thought reasoning (Nori et al., 2023) aligned with radiological protocols, (2) Curated few-shot examples for fracture patterns, (3) Strict "explicitly stated observations" scope, and (4) Comprehensive confidence scoring and audit trails.

## 2.2. Pseudo-Labelling

We utilize YOLOv7 (Wang et al., 2023) pretrained on GRAZPEDWRI-DX dataset (Nagy et al., 2022) for initial labeling. To address the domain shift between datasets (Figure 1b) that limits accuracy to 76%, we developed two strategies: 1. Single fracture focus: Prioritizing cases with isolated fractures to reduce location mismatches. 2. VLM verification: Using vision-language models to verify anatomical correctness of detected regions (Al Mohamad et al., 2025).

2.2.1. MATCHING YOLO LABELS

The simplest method to match YOLO labels with radiological fractures involves counting the number of fractures mentioned in the report and checking if YOLO has labeled the same number of fractures in the corresponding images. However, several challenges can arise: 1) YOLO labels may be misplaced, such as incorrectly marking phalanges instead of actual fractures (as shown in Figure 1a), 2) not all images from the examination may display all fractures. To address these issues, we propose two strategies: 1) **Single fractures**: We focus on cases with only a single fracture. This approach reduces the likelihood of mismatches between the number of classes and their correct locations. 2) **Vision-Language**

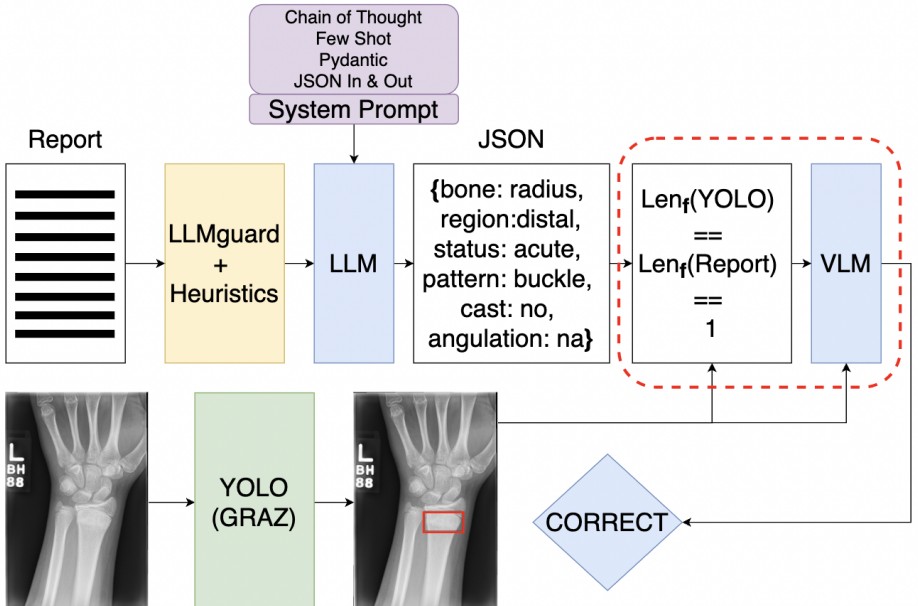

Figure 2: Overview of the proposed method. Radiological report is passed through LLM-guard framework to remove all PHI and unncessary exam data, followed by API call to Sonnet 3.5 LLM with strict JSON input/output rules, chain-of-thought reasoning over clinical radiological examination guidelines, few shot examples and Pydantic control over datatypes. The resulting JSON is checked for consistency by 1. number of fractures 2. visual cues alignment via VLM. The full pipeline is fully automated and requires zero radiologist intervention. Resulting correct samples are used to finetune YOLO model.

**Model (VLM)**: Although non-medical LLMs are not specifically trained to detect the location, class, or severity of fractures, they have been shown to be surprisingly effective at understanding general anatomy in medical images (Al Mohamad et al., 2025). To enhance our system's accuracy, we can encode the YOLO-labeled image along with the clearly visible bounding box into base64 format and query the VLM to determine whether a specific area of the image is highlighted. This provides an additional "guardrail" for our system, improving label matching with actual fractures.

### 2.3. Data Collection

An extensive dataset of 2939 clinical pediatric X-ray examinations was retrospectively acquired from Boston Children's Hospital radiology department under an appropriate IRB protocol. The dataset included clinical radiology reports spanning January 2013 through August 2024. Each examination consisted of multiple image projections following standard radiological protocols, including anteroposterior, lateral, oblique and scaphoid views when clinically indicated.

### 2.3.1. ANONYMIZATION PROCEDURE

A multi-stage anonymization protocol was implemented to ensure data privacy and compliance before conducting LLM operations. First, a rule-based system was employed to remove explicit identifiers, including dates, age, names, gender, medical record numbers and other tags related to protected health information. Additionally, DICOM metadata was stripped of patient and exam specific identifiers. Following this initial step, LLMguard framework (Protect AI, 2024) was used as an extra verification layer to detect any remaining nominal identifiers.

## 2.4. Annotation

### 2.4.1. DATA

A stratified random sample of 300 reports was selected for detailed validation, with stratification based on fracture types and anatomical locations. A board-certified radiologist independently reviewed 300 selected reports and corresponding images. To facilitate this process, a labeling interface for deriving reference bounding boxes was constructed using the Flask library (Pallets, 2024). This interface allowed the radiologist to access all images associated with each report, view the anonymized report itself, and create bounding box labels. The reference dataset was then divided into three parts: 214 images for fine-tuning the reference method, 16 images for validation, and 70 images for testing various methods, including the proposed method and comparison methods.

Table 1: Distribution of Wrist Fractures and Their Characteristics

| Characteristic | Count | Percentage (%) |
|---|---|---|
| *Anatomical Location* | | |
| Radius | 934 | 86.2 |
| Scaphoid | 94 | 8.7 |
| Ulna | 41 | 3.8 |
| Other | 14 | 1.3 |
| *Fracture Region* | | |
| Distal | 717 | 66.2 |
| Metaphysis | 267 | 24.7 |
| Other | 99 | 9.1 |
| *Fracture Pattern* | | |
| Transverse | 230 | 21.2 |
| Torus | 149 | 13.8 |
| Simple | 74 | 6.8 |
| Oblique | 26 | 2.4 |
| Comminuted | 23 | 2.1 |
| *Healing Stage* | | |
| Healing | 543 | 50.1 |
| Acute | 261 | 24.1 |
| Early Healing | 101 | 9.3 |
| Other | 178 | 16.5 |

## 2.5. YOLO

### 2.5.1. Architecture

The standard width architecture included 5-stage backbone with 32 to 1024 channels for feature extractiona and a detection head with SPPCSPC module. 9 anchor boxes were used across 3 scales (P3/8, P4/16, P5/32), and RepConv blocks were employed for final feature refinement before IDetect-based prediction.

### 2.5.2. Training

The training pipeline was based on a multi-scale strategy ($640\times640$ base resolution) and a batch size of 16. SGD optimizer was set with learning rate of 0.001, momentum of 0.937, and weight decay of 0.0001. A one-cycle policy with cosine annealing (final LR factor 0.2) and 3-epoch warmup period was used for scheduling.

Loss function was tuned tuned for X-ray image detection: classification loss weight was reduced (0.1), objectness loss weight was increased (0.8), and IoU threshold was lowered (0.15) to accommodate imprecise fracture region boundaries. Augmentation was added as per X-ray data needs with geometric transformations (rotation $\pm10°$, scale $\pm0.5$, shear $\pm0.2$), brightness variation (HSV $\pm0.3$, to simulate exposure variations). Complex augmentations (mosaic, mixup) were disabled to preserve anatomical coherence. Training was conducted on NVIDIA A100 GPUs. The training process incorporated a balanced sampling strategy ensuring representation of both common and rare fracture types within each batch. There were two principal types of dataset. The reference dataset included 300 radiologist labels, as described in the section above. 300 images were split into 214 images for finetuning, 16 images for validation and 70 images for testing. Training involved 3 datasets: a) manually annotated dataset by expert radiologist (230 images) b) small pseudo-annotated dataset (300 images) c) large pseudo-annotated dataset (5700 images).

## 2.6. Evaluation

### 2.6.1. Quantitative Evaluation

The extracted structured data was evaluated using standard metrics. Analysis of fracture characteristics was measured for precision, recall and F1 scores.

## 3. Results

Our pseudo-blind finetuning approach with the large dataset (5700 images) achieved superior performance across key metrics compared to baseline models. The method improved mean Average Precision (mAP) from 0.789 (Graz pretrained) to 0.822, while maintaining high confidence thresholds (0.961) and recall (0.902). Notably, this performance matches or exceeds that of manual expert annotation (mAP: 0.805), particularly in challenging cases involving multiple fracture patterns or growth plate proximity. Analysis of the LLM-based extraction system showed 93% schema compliance with a processing time of 2.3 seconds per report. Error analysis revealed challenges primarily in complex fracture patterns (15% error rate) and ambiguous language interpretation (12% error rate), suggesting areas for future improvement.

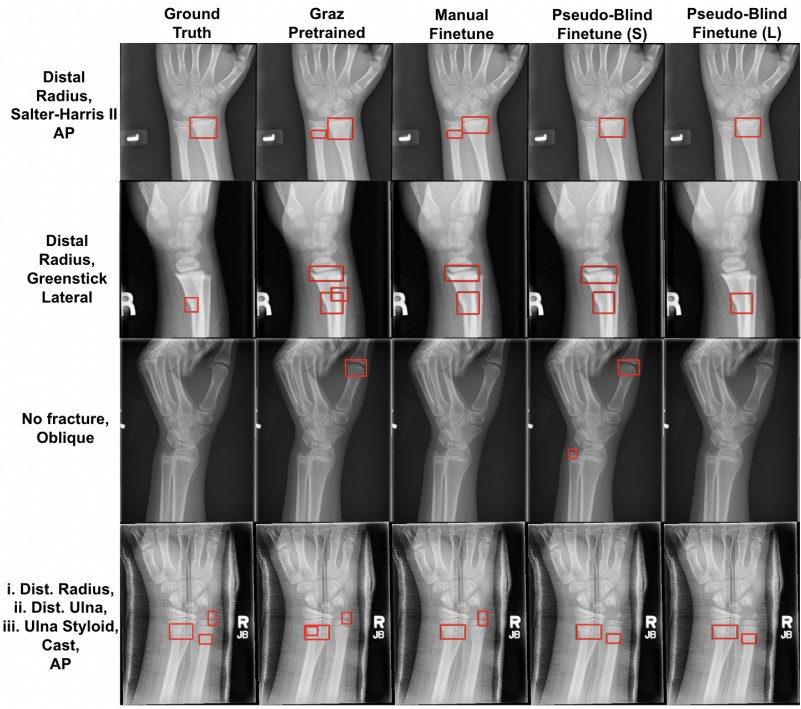

Figure 3: Comparison of YOLO trained methods against radiologist marked ground truth on variety of clinically relevant cases from different fracture regions & patterns, image views and conditions (e.g. cast). Pseudo-blind finetune (Ours) performs best with the large training dataset and is either better (rows 1&2) or on par (row 3) with Manual Finetune. In Row 4, neither model succeeds entirely - Pseudo-Blind Finetunes correctly predict two distal fractures, while the Manually Finetuned model and the Graz Pretrained model correctly identify the styloid and distal radius.

## 4. Discussion and Conclusions

We demonstrated that LLM-powered information extraction from radiology reports can effectively enhance fracture detection models, achieving superior performance (mAP 0.822) compared to manual annotation approaches without requiring additional expert labeling. Our pseudo-blind finetuning method particularly excelled with pediatric cases where growth plates typically challenge traditional detection systems.

While the approach showed promise, limitations emerged with complex fracture patterns (15% error rate) and report language ambiguity (12% error rate). The model exhibited reduced performance on cast-containing images due to radio-opacity interference with bone structure visualization. Despite balanced sampling implementation, class distribution skewness, particularly in rare fracture patterns (e.g., styloid fractures at 3.8%), impacted model performance on underrepresented classes.

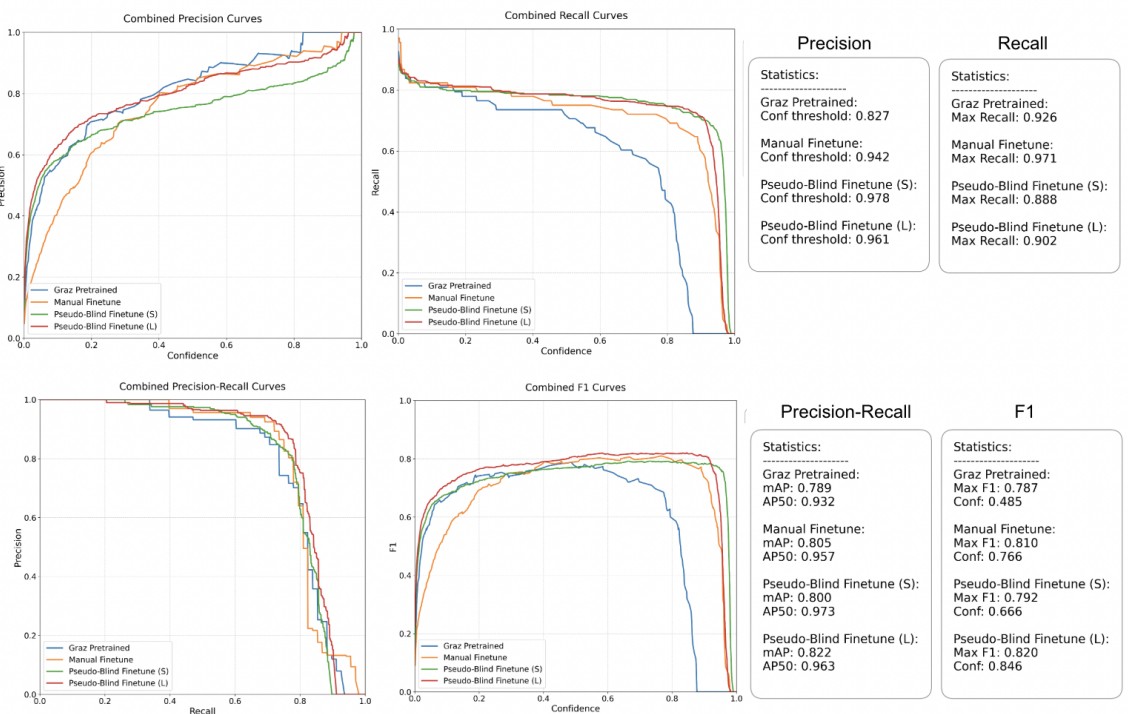

Figure 4: Comparison of model performance metrics across different training approaches. Combined Precision (top left) and Recall curves (top right) show confidence thresholds versus respective metrics. Precision-Recall curves (bottom left) and F1 scores (bottom right) demonstrate overall detection performance. The Pseudo-Blind Finetune with large dataset (L) achieves superior performance with mAP of 0.822 and maximum F1 score of 0.820, outperforming both the Graz Pretrained baseline (mAP: 0.789) and Manual Finetune approach (mAP: 0.805). Statistics panels show detailed performance metrics for each model variant, with Pseudo-Blind Finetune (L) demonstrating the highest confidence threshold (0.961) and competitive recall (0.902) compared to manual annotation.

Future development should focus on specialized augmentation for cast-present cases and advanced sampling techniques like dynamic hard negative mining to address class imbalance while preserving model stability. Additional areas for improvement include enhanced robustness to report variability. This study establishes the feasibility of using LLM-extracted knowledge to reduce annotation burden while maintaining high diagnostic accuracy in clinical settings.

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
