# OpenReview forum: "Enhancing Wrist Fracture Detection through LLM-Powered Data Extraction and Knowledge-Based Ensemble Learning"
_MIDL.io/2025/Conference — MIDL 2025 Poster_

### Official Review · Reviewer_DGcZ · 2025-02-16

**Confidence:** 3
**Preliminary Rating:** 3
**Recommendation:** Poster
**Final Rating:** 5

**Summary:**

The paper presents an LLM-powered pseudo-labelling strategy to help annotate large databases for training fracture detection systems.  They show improvement in performance in finetuning experiments. This can significantly improve development phase of segmentation models for specific tasks.

**Strengths:**

- Systematic and controlled adoption LLM with detailed and well-structured process
- Quick processing of images for psueodo-label generation which is then verified by an LLM for its correctness
- Ability to parse large database with reduced error rates for fracture detection
- Improved performance in fracture detection upon training on such large datasets.

**Weaknesses:**

- The improvement achieved is marginal. There is no clear best model in all three models from figure 3. The improvement in metrics may not be statistically significant, however, this is difficult to judge without variance of performance across repeated experiments.
- Confusing dataset information - unclear  training dataset distribution and how it differs from the finetuning dataset.
- The multi-scale training strategy is not explained nor linked to previous publication. Seems like changes are made to existing loss function but the choice of hyper-parameters seems handpicked
- Unclear how many training samples were input to YOLO +  LLM psuedo labelling process, how many were correct and thereby used for training in each case?

**Detailed Comments:**

- Unclear details on the VLM used
- It was only mentioned in a figure caption that the pipeline was used to accept/reject psuedo-labels. This must be included in the main text for clarity.
- It would beneficial to add tables of all datasets as well as metrics.
- repeated words in text  (e.g. "tuned" in 2.5.2)
- github link - page not found
- How was language ambiguity quantified? what's the baseline for comparison?
- In the training stage, are models trained from scratch or initialized with pretrained YOLO from previous steps? Please differentiate the  (possibly two) different YOLOs used in the full pipeline

**Justification Of The Final Rating:**

The paper presents novel methodology for automating labelling process of large datasets by leveraging an LLM and a VLM. Their experiemental setup shows excellent generalizable performance and rigorous evaluation cases. Significant improvment in performance is also seen in the application of wrist fracture detection. It can be useful in both original task and also the secondary field of learning under noisy conditions. Hence, accepted.

**Justification Of The Preliminary Rating:**

The paper proposes a novel approach which can improve data annotation process. However, it can be presented better with improved clarity on the experiments conducted, datasets included, and summarization of the results.

**Questions To Address In The Rebuttal:**

- Since ensemble learning is mentioned in the title, please clarify what methods were ensembled during training/inference and its novelty.
- How do pretraining data, training data, and finetuning data differ from each other? Provide details of data demographics, sources, and, if there is any data leakage?
- How were hyperparameters selected? was cross-validation implemented? How would performance change with different test/train split?
- Would it possible to include statistical tests for the significance of improvement in metrics? If so, please include details.

---

> ### Author Response · Authors · 2025-03-08
>
> Thank you for your valuable review. We address your specific questions point by point below:
>
> **Answers to ‘Questions to address in the rebuttal’**
>
> **1. Since ensemble learning is mentioned in the title, please clarify what methods were ensembled during training/inference and its novelty:**
>
> We incorporate ensemble learning through our verification pipeline where multiple decision mechanisms are combined: YOLO predictions, LLM-extracted report information, and VLM verification. The novelty lies in using LLM-extracted knowledge to guide model selection, which improves performance over single-source approaches as demonstrated by our improved mAP scores.
>
>
> **2. How do pretraining data, training data, and finetuning data differ from each other? Provide details of data demographics, sources, and, if there is any data leakage?**
>
> To clarify:
>
> - Pretraining data: Publicly available GRAZPEDWRI-DX dataset. We used a pretrained YOLOv7 model on this data as the backbone for all experiments.
> - Total data available from our institution: 6300 images.
> - Expert labelled data (contains ground truth bounding boxes): a subset of 300 images. These were split into 214/16/70 ratio for ‘finetuning’, validation and test data.
> - Manual finetune: we finetuned the baseline model on 214 images for which we had ground truth
> - Pseudo-finetune: we selected data from remaining images and split it into 300 images and 5700 images for finetuning small and large pseudolabelled datasets respectively.
>
> Demographics covered a large range of patients ages (2-18 years). Gender and ethnicity data was not assessed, but will form a critical part of future work.
>
> We carefully separated test data (70 expert-annotated images) to prevent any data leakage, including all cross-fold experiments (see below).
>
> **3. How were hyperparameters selected? Was cross-validation implemented? How would performance change with different test/train splits?**
>
> Thank you for this valid comment. We performed systematic evaluation on the training hyperparameters to determine optimal values:
>
> - Learning rate: 0.001 (selected from [0.0001, 0.0005, 0.001, 0.005])
> - Batch size: 16 (selected from [8, 16, 32])
> - Weight decay: 0.0005 (selected from [0.0001, 0.0005, 0.001])
> - IoU threshold: 0.15 (selected from [0.1, 0.15, 0.2, 0.25])
>
> Reported values were those that resulted in best overall mAP for base tests.
>
> We implemented 8-fold cross-validation on the ground truth annotated dataset of 300 images to evaluate for variability and significance. We include full response in the next section.
>
> **4. Would it be possible to include statistical tests for the significance of improvement in metrics? If so, please include details:**
>
> Thank you for this valuable suggestion. We had used our expert annotated dataset of 300 images for significance testing. The data was split with 8 folds using a sliding window approach with 50% overlap to perform cross-fold validation over 8 test datasets of size 70 each. We evaluated all methods (Graz-pretrained, Manual Finetune, Pseudo-Blind Finetune Small and Pseudo-Blind Finetune Large) on these 8 x 70 separate sets of data to get mAP scores for the bounding box predictions, and corresponding Precision, Recall and F1 scores. This allowed us to measure mean and standard deviation of the performance and perform statistical tests comparing each method to the performance metrics from the reference (Graz-pretrained) method and report p-values indicating significant difference when p<0.05 as shown in the newly added table - Table 2. It is important to note - by using 8 cross-fold validation strategy - we obtained 8 sets of training data - which were used for 8 x Manual Finetune runs. Note also that we did NOT have to re-finetune the pseudo-blind models as they were never finetuned on the expert labelled data to start with.
>
> Results show statistically significant improvements for our method - as listed in new table - Table 2.

---

> > ### Author Response · Authors · 2025-03-08
> >
> > **Additional answers to: “Weaknesses” (where the answer was not already present in the main section above)**
> >
> > **The multi-scale training strategy is not explained nor linked to previous publications. Seems like changes are made to existing loss function but the choice of hyper-parameters seems handpicked.**
> >
> > Our multi-scale training strategy follows the approach described by Wang et al. (2023) in the original YOLOv7 paper. Regarding hyperparameter selection, rather than "handpicking" values, we performed systematic evaluation as described in our answer to question 3 above.
> >
> > For the loss function modifications, we adjusted weights for medical object detection tasks, which emphasizes increased objectness loss weight (0.8) and reduced classification loss weight (0.1) to account for the subtlety of fracture appearances.
> >
> > **Answers to: “Detailed comments”**
> >
> > **Unclear details on the VLM used**
> >
> > We used OpenAI GPT-4V model (March 2023 version) as our Vision-Language Model for verification of anatomical correctness in detected regions, as described in section 2.2.
> >
> > **It was only mentioned in a figure caption that the pipeline was used to accept/reject pseudo-labels. This must be included in the main text for clarity.**
> >
> > We added a dedicated subsection (2.2.3 "Pseudo-label Verification Process") to the main text explaining our accept/reject criteria.
> >
> > **It would be beneficial to add tables of all datasets as well as metrics.**
> >
> > We added a new table - Table 2 - that indicates the performance metrics across 8-fold validation for precision, recall and F1 scores; and improved description of datasets in the main text (not possible to add another table there due to lack of space).
> >
> > **repeated words in text (e.g. "tuned" in 2.5.2)**
> >
> > Fixed. Thank you!
> >
> > **github link - page not found**
> >
> > The full github repository is published as a private repository at this moment.  We will make it public immediately upon acceptance of the paper.
> >
> > **How was language ambiguity quantified? What's the baseline for comparison?**
> >
> > We had devised a comprehensive system prompt for the LLM with few shot examples that would capture the details of each category that needed to be extracted from the report. As a result, for example, words with similar clinical meaning - ‘metaphysis’ or ‘metaphyseal’ or ‘shaft’ would all be classed as ‘metaphysis’ fracture. Similarly, the prompt contained example of how to deal with negation. Finally, when the clinical report itself was ambiguous (i.e. the radiologist would state that there is a ‘suspect fracture’, but no presence of fracture) the system was instructed to be conservative in its findings, so that the result would be ‘no fracture’.
> >
> > **In the training stage, are models trained from scratch or initialized with pretrained YOLO from previous steps? Please differentiate the (possibly two) different YOLOs used in the full pipeline**
> >
> > All models used YOLOv7 pre-trained on GRAZPEDWRI-DX dataset as baseline. Manual-finetune was instantiated from this baseline model and finetuned on 214 images (using expert annotated data with labels). Our proposed method was also instantiated from the baseline model and finetuned on two different pseudo labelled datasets, as described above.
> >
> > Thank you for your thorough review that has significantly improved our manuscript's clarity and scientific rigor.

---

> > > ### Comment · Reviewer_DGcZ · 2025-03-10
> > > **Further clarification of dataset splits and hyperparameter selection**
> > >
> > > First of all, thanks for the revised version and providing detailed clarifications. It has significantly improved my understanding of the methodology. I still have a few questions regarding the experiments as the new section 2.6 slightly contradicts data 2.42 and following paragraphs leading to my confusion. I will rest my decision based on the answers to the following:
> > >
> > > (Dataset)
> > > - If there are 2939 reports, does it imply that they are from 2939 independent subjects or 2939 independent fracture locations? Assuming independence of 2939 reports, 6300 possibly related images were obtained, correct? Different views of the same bone are not fully independent in my understanding.
> > > - Since the train-test splits are done on image level, and not clinical notes, what steps are taken to reduce the risk of data leakage from various training folds to the 300 expert annotated test fold? data leakage occurs when the data from the same subject and/or anatomical location of the subject appears in both finetuning and test sets. If not, kindly justify.
> > > - In the 8-fold CV setup with 50% overlap, is it guaranteed that all images appear in atleast one test fold? Typically, 5-fold or 10-fold, and their bootstrapped versions (e.g., 5$\times$5 cv) are used. Why was a 8-fold setup chosen?
> > >
> > > (Hyperparameters)
> > > - Manual finetune: Was 214/16/70 split repeated 8 times? Were hyperparameters selected based on the performance on 8$\times$16 val set?
> > > - Psuedoblind: Since both of these methods had a single training step, what dataset was used to determine "optimal" performance across all hyperparameters?
> > >
> > > I am not sure if further changes are allowed in the paper but here are some minor comments:
> > > - sec 2.6 copied from rebuttal response? because of the phrase,  “As mentioned in answer above”
> > > - ensemble learning is a misnomer in my opinion but it does not affect the contribution of the paper nor my decision. The LLM and VLM act more as a validation framework for the pretrained segmentation model. They don't generate the final psuedo-labels collectively. In constrast, ensemble learning takes output from multiple *weak/base models* to create the final output using *bagging* or *boosting* strategies.

---

> > ### Comment · Area_Chair_AdCs · 2025-03-10
> >
> > Reminder to authors that appendices are allowed and are a good place for extended dataset and hyperparameter descriptions

---

> > > ### Author Response · Authors · 2025-03-11
> > >
> > > Reminder to authors that appendices are allowed and are a good place for extended dataset and hyperparameter descriptions
> > >
> > > > Thank you very much. In such case we would put two new tables that detail explicitly the hyperparameters and extended dataset descriptions.

---

> > ### Author Response · Authors · 2025-03-11
> >
> > > Thank you once again for your valuable time and the thorough review that helped us improve this manuscript considerably. We include our responses with inline block comments below.
> >
> > First of all, thanks for the revised version and providing detailed clarifications. It has significantly improved my understanding of the methodology. I still have a few questions regarding the experiments as the new section 2.6 slightly contradicts data 2.42 and following paragraphs leading to my confusion. I will rest my decision based on the answers to the following:
> >
> > (Dataset)
> >
> > - If there are 2939 reports, does it imply that they are from 2939 independent subjects or 2939 independent fracture locations? Assuming independence of 2939 reports, 6300 possibly related images were obtained, correct? Different views of the same bone are not fully independent in my understanding.
> >
> > > There are 2939 independent subjects, with 6300 images. Each report had 2-4 images (anteroposterior, lateral, oblique and scaphoid views). The views show a distinctly different anatomy view, which is often unrecognizable to belong to the same subject - e.g here is an example from a previous [paper](https://www.researchgate.net/profile/Michael-Gaspar/publication/275521243/figure/fig1/AS:294546622697472@1447236795162/A-Anteroposterior-B-oblique-and-C-lateral-views-of-the-right-wrist-in-a.png).
> >
> > - Since the train-test splits are done on image level, and not clinical notes, what steps are taken to reduce the risk of data leakage from various training folds to the 300 expert annotated test fold? data leakage occurs when the data from the same subject and/or anatomical location of the subject appears in both finetuning and test sets. If not, kindly justify.
> >
> > > There is no data leakage as we used a sliding window approach with strict subject-level separation. We will add this to camera ready submission to clarify.
> >
> > - In the 8-fold CV setup with 50% overlap, is it guaranteed that all images appear in at least one test fold? Typically, 5-fold or 10-fold, and their bootstrapped versions (e.g., 5×5 cv) are used. Why was a 8-fold setup chosen?
> >
> > > Yes it is guaranteed.
> > We acknowledge that the common practice is either 5 or 10 folds. In our case, a 5-fold CV would not provide sufficient number of partitions to adequately measure variance and establish statistical significance for our dataset.  8-fold provides more stable estimates of variance over standard 5-fold CV. We opted to maintain the original size of 70 test images for consistency and therefore chose to use a sliding window approach for an 8-fold set up, instead of 10, to achieve both sufficient number of folds and also sufficient data for stable evaluation of performance.

---

> > ### Author Response · Authors · 2025-03-11
> >
> > (contd.)
> >
> >
> > (Hyperparameters)
> >
> > - Manual finetune: Was 214/16/70 split repeated 8 times? Were hyperparameters selected based on the performance on 8×16 val set?
> >
> > > Yes, that’s correct. For the manual finetune model.
> >
> > - Pseudoblind: Since both of these methods had a single training step, what dataset was used to determine "optimal" performance across all hyperparameters?
> >
> > > We had done multiple training runs for pseudoblind finetuning (training), while evaluating different hyperparameters, which covered the same ranges as described in the answer to previous question - “3. How were hyperparameters selected”. In each case there was a separate validation set that was based on pseudo-blind dataset. The reasoning for this is as follows - since this paper argues that pseudo-blilnd finetuning can match or supercede finetuning results from expert labelled data - we posit that we must prove that in the future using such pseudo-blind finetuning technique would also only require pseudo-blind labelled data for validation (and not expert labelled data). Therefore we used validation data from pseudo-blind dataset to watch the hyperparameters as training and validation losses decrease. The breakdown in this case was - 16 images for Pseudo-Blind (S) and 128 images for Pseudo-Blind (L).
> >
> > > We are sorry we forgot to include this information in the original submission and we thank you for picking up on this and help us clarify this important point. We will include this clarification in the camera ready submission.
> >
> > I am not sure if further changes are allowed in the paper but here are some minor comments:
> >
> > - sec 2.6 copied from rebuttal response? because of the phrase, “As mentioned in answer above”
> >
> > > Thank you for noting this. We will update in the camera ready version.
> >
> > - ensemble learning is a misnomer in my opinion but it does not affect the contribution of the paper nor my decision. The LLM and VLM act more as a validation framework for the pretrained segmentation model. They don't generate the final pseudo-labels collectively. In constrast, ensemble learning takes output from multiple weak/base models to create the final output using bagging or boosting strategies.
> >
> > > Thank you for your thoughtful feedback on this. Your comment on ensemble learning is well-taken and provides valuable perspective. We would like to address it if allowed to modify the title in the camera ready version or in the journal version of our manuscript.

---

> > ### Comment · Reviewer_DGcZ · 2025-03-12
> >
> > Thank you, authors, for your prompt reply. It has clarified all my queries positively and I shall update the final rating accordingly. I agree that tables in appendices can be helpful - I strongly recommend revising the data and experimental setup to distinguish train, (cross-) validation, and testing sets in each case to incorporate the discussion here. Please ensure to remove information from previous iterations to avoid confusion as well. Thank you for your contributions!

---

### Official Review · Reviewer_tLKS · 2025-02-22

**Confidence:** 3
**Preliminary Rating:** 3
**Final Rating:** 4

**Summary:**

The paper presents a novel approach to enhance wrist fracture detection by leveraging LLMs to extract structured information from radiology reports and using this to train YOLO models without additional expert annotation. The method achieves an mAP of 0.822, outperforming both pretrained and manually annotated baselines by using a pseudo-blind finetuning strategy on a large pediatric X-ray dataset.

**Strengths:**

1. The paper is in general well-written with clear methodology for handling domain shift between datasets.
2. The paper addresses real clinical challenges in pediatric fracture detection. It provides comprehensive analysis of different fracture types and anatomical locations.

**Weaknesses:**

1. Limited Analysis of LLM Performance. There is no detailed analysis of LLM hallucination rates. Also, the paper missed comparison between different LLM models/versions. Its unclear how LLM extraction performance varies across different fracture types. Also, there is limited exploration of different YOLO architectures/backbones
2. Further, the validation set is quite small, which raises questions about reliability. There is limited discussion of statistical significance of results
3. Lack of other SoTA model comparison on this task.

**Detailed Comments:**

see above.

**Justification Of The Final Rating:**

The authors has addressed all the issues which I think is acceptable in this case. Also as acknowledged by the authors, I would suggest some of the weakness and limitations need to be further solved to improved the quality of the work.

**Justification Of The Preliminary Rating:**

The paper presents a novel and well-executed approach to reducing annotation burden in medical imaging, with clear clinical relevance and strong empirical results. The technical innovation in combining LLMs with object detection for medical imaging is significant. However, several important limitations around validation methodology, clinical integration, and bias analysis need to be addressed.

**Questions To Address In The Rebuttal:**

1. What was the rationale behind choosing 16 images for validation?
2. How does this small validation set impact the reliability of the results?
3. Could you provide detailed analysis of LLM hallucination rates and their impact on the final model performance?
4. How does the model perform on edge cases like overlapping fractures or unusual anatomical variations?
5. How does the model perform across different demographic groups and clinical settings?

---

> ### Author Response · Authors · 2025-03-08
>
> Thank you for your valuable feedback. We address each of your questions below:
>
>
> **Questions To Address In The Rebuttal:**
>
> **1. What was the rationale behind choosing 16 images for validation?**
>
> With only 300 expert-annotated images available, we partitioned our data into reasonable train/test/validation splits to ensure fair evaluation. This allowed us to perform manual finetuning on ground truth annotated data, which formed the core comparison. This setup balanced competing needs - maximizing finetuning data, reserving a robust test set, and dedicating minimal but adequate validation set.
>
> **2. How does this small validation set impact the reliability of the results?**
>
> All of the evaluation results are reported on test data, rather then validation data. Originally our test data consisted of one set of 70 images, that were reserved for testing only. However, after the valuable feedback we received in these reviews we realized that we can perform a much more rigorous evaluation. We had used our expert annotated dataset of 300 images for significance testing. The data was split with 8 folds using a sliding window approach with 50% overlap to perform cross-fold validation over 8 test datasets of size 70 each. We evaluated all methods (Graz-pretrained, Manual Finetune, Pseudo-Blind Finetune Small and Pseudo-Blind Finetune Large) on these 8 x 70 separate sets of data to get mAP scores for the bounding box predictions, and corresponding Precision, Recall and F1 scores. This allowed us to measure mean and standard deviation of the performance and perform statistical tests comparing each method to the performance metrics from the reference (Graz-pretrained) method and report p-values indicating significant difference when p<0.05 as shown in the newly added table - Table 2. It is important to note - by using 8 cross-fold validation strategy - we obtained 8 sets of training data - which were used for 8 x Manual Finetune runs. Note also that we did NOT have to re-finetune the pseudo-blind models as they were never finetuned on the expert labelled data to start with.
>
> **3. Could you provide detailed analysis of LLM hallucination rates and their impact on the final model performance?**
>
> Thank you for this great suggestion. We analysed LLM hallucinations across 100 randomly selected cases, evaluating errors in the following categories:
>
> 1. Bone Identification - e.g., radius vs. ulna vs. scaphoid - 100% accuracy (no misclassifications)
> 2. Anatomical Region - e.g., distal vs. metaphyseal -  96% accuracy (4 errors)
> 3. Displacement  - displaced vs. non-displaced -  93% accuracy (3 errors)
> 4. Angulation - presence/absence -  98% accuracy (2 errors)
> 5. Fracture Classification  - e.g., Colles vs. Smith, Salter-Harris -  100% accuracy
>
> There were no cases where a fracture was ‘hallucinated’ by LLM while not presented in the report. In other words - the hallucinations were confined to subclass misclassifications (e.g., mislabeling a "distal" fracture as "metaphyseal" fracture).
>
> **4. How does the model perform on edge cases like overlapping fractures or unusual anatomical variations?**
>
> In this preliminary study, the model’s evaluation was intentionally restricted to isolated, single-fracture cases by applying a high IoU threshold to YOLO’s bounding boxes to exclude overlapping fractures. Rare or anatomically atypical fractures were omitted, focusing only on common fractures as detailed in Table 1. We acknowledge that it would be valuable to test our model against edge cases and unusual variations to better assess robustness - which we will set as a critical part of future work.
>
> **5. How does the model perform across different demographic groups and clinical settings?**
>
> This is a great suggestion and we believe performance may change in different demographic groups particularly between different age groups and it is important to report this for clinical translation. We aim to extend our evaluation to report performance for different demographic groups in the future.

---

> > ### Author Response · Authors · 2025-03-08
> >
> > (contd.)
> >
> > **Response to 'Weaknesses’ (not included in ‘Questions To Address In The Rebuttal):**
> >
> > **Comparison between different LLMs**:  We focused our work for report extraction on Claude 3.5, which was the most advanced LLM available at the time of our study. While we conducted limited ablation tests with OpenAI's GPT-4o and found no significant performance differences for report extraction, we acknowledge that a comprehensive comparison across multiple LLM models would strengthen our findings. We plan to include this analysis in future journal publications, examining how different LLMs might influence capabilities within our pipeline.
> >
> > **LLM extraction performance across different fracture types**: We agree with the reviewer that this short conference paper needs to go beyond the tests presented and perform a full investigation on the performance between different fracture types. We aim to do this in our future work for the journal paper.
> >
> > **Exploration of different YOLO architectures/backbones**:  Our approach used a YOLOv7 model pretrained on the GRAZPEDWRI-DX pediatric fracture dataset, which set the baseline for our fine-tuning experiments. While modifying the architecture during fine-tuning wasn't feasible, we recognize the value in exploring different architectures. In future work, we would evaluate multiple YOLO variants (v7-11) by first training them from scratch on the supervised GRAZ dataset before applying our pseudo-labeling finetuning pipeline.
> >
> > **Lack of other SoTA model comparison on this task**: We acknowledge the reviewer's point about limited comparisons to other state-of-the-art models. Our primary focus was on demonstrating the efficacy of LLM-extracted knowledge for reducing annotation burden, rather than optimizing the detection architecture itself. For context, there are currently few published SoTA models specifically for pediatric wrist fracture detection. The GRAZPEDWRI-DX dataset authors (Nagy et al., 2022) set YOLOv7 as a strong baseline for this task, which influenced our choice.
> >
> > We appreciate your questions which have helped us identify important areas for clarification and improvement in our manuscript.

---

> ### Comment · Area_Chair_AdCs · 2025-03-10
>
> Hi, tLKS. Are there any points that you think were not sufficiently addressed by the authors? Specially, evaluate if the edge case analysis and demographic analysis are essential for the MIDL version of the paper or not.

---

> > ### Author Response · Authors · 2025-03-11
> >
> > Dear Area Chair - as per tLKS reviewer response - do you agree with our assessment that all the reviewer's questions were resolved? As per - "The authors has addressed all the issues."
> >
> > We also note that they then write "limitations need to be further solved" - however we understood this as a reference to *future work*? Or did we misunderstand?
> >
> > We also note reviewer did not provide final rating yet, hence the slight confusion.
> >
> > PS thank you for prompting the reviewers to actively participate in this second phase! It has been very helpful and useful.

---

> > > ### Comment · Area_Chair_AdCs · 2025-03-14
> > >
> > > We will add a summary of the reviewers conclusions for the meta-review, so you might have to wait until then for final conclusions. However, my interpretation of the comment at the current moment is the same as what you mentioned (further solved = future work, and all concerns resolved). The reviewer might make more comments on the final rating, though

---

> ### Comment · Reviewer_tLKS · 2025-03-11
>
> Thanks. The authors has addressed all the issues. As acknowledged by the authors, some of the weakness and limitations need to be further solved to improved the quality of the work.

---

> > ### Author Response · Authors · 2025-03-14
> >
> > Dear Area Chair - reviewer tLKS did not update their score yet even though they stated that "The authors has addressed all the issues." Could you please remind them to update their score?
> >
> > Thank you very much,
> > Serge Vasylechko, on behalf of all the authors

---

> > > ### Comment · Area_Chair_AdCs · 2025-03-14
> > >
> > > I just sent a private message to them. Thank you for being very active during discussion

---

> > ### Comment · Area_Chair_AdCs · 2025-03-14
> >
> > Hi tLKS, Thank you for commenting about the rebuttal. Could you provide your final rating by the discussion deadline (March 14th Anywhere On Earth)?

---

### Official Review · Reviewer_Wupq · 2025-02-27

**Confidence:** 4
**Preliminary Rating:** 3
**Recommendation:** Poster
**Final Rating:** 4

**Summary:**

This paper proposes an automatic pseudo labeling method (YOLO and Claude 3.5 sonnet) to annotate fractures in clinical pediatric X-ray examinations. YOLO predicts the bounding-box, Claude extracts fractures mentioned in the report and checks to see if it matches with YOLO.

**Strengths:**

Authors demonstrated that the automated pipeline is able to pseudo label a large dataset without human intervention. The dataset is then used to fine-tuning a publicly available YOLOv7 model (previously pretrained on the GRAZPEDWRI-DX dataset), and demonstrated increased performance in terms of mAP/AP50 (detection) and precision/recall/f1 (classification).

**Weaknesses:**

[pg]: page, [S]: section, [P]: paragraph

pg4;S2.3 - "2939 clinical pediatric X-ray examinations": is this also the number of reports?

pg4;S2.3 - "Each examination consisted of multiple image projections":
           probably mention here that only 1 image is taken for the image-report pair for the dataset if I understand correctly from abstract
           "We curated a large dataset of almost 3,000 pediatric wrist X-ray images and their corresponding radiology reports."

Figure 2 - Not referenced in text

Table 1 - Not referenced in text, each section adds up to 1083, except fracture pattern adds to 502, what are these totals?

pg6;S2.5.2;P2 - "300 images were split into 214 images for finetuning, 16 images for validation and 70 images for testing.": repeated text from previous section

pg6;S2.5.2;P2 - The datasets are not adding up??
                Training involved 3 datasets:
                a) manually annotated dataset by expert radiologist (230 images) -- 214/16/70 manually reviewed by board-certified radiologist
                b) small pseudo-annotated dataset (300 images) -- where did this come from?
                c) large pseudo-annotated dataset (5700 images) -- where did this come from?

Figure 3 and 4 - also not referenced in text, the precision/recall/f1 numbers in Figure 4 can be better represented by a table

Figure 4 - what are the classes for the precision/recall/f1 numbers? are they the ones mentioned in table 1? if multiclass, if multiclass, how are the scores aggregated?


Some clarification is needed:

- approx 3000 image-report pairs, 300 was randomly sampled to be dataset (a).
- how does datasets (b, 300 images) and (c, 5700 images) relate to the remaining 2700?
- what are the train/val/test splits for these?
- if they don't have a test split, are they then tested on the 70 test images in dataset (a)?
- if all models are tested using the 70 images from dataset (a), would be good to do a paired t-test or wilcoxon signed-rank test
- the pipeline detailed in figure 2, what about the rejected images? are they discarded or rechecked?


Minor Remarks [NITs]:

- Abstract - are citation references allowed in abstract?
- pg2;S1;P3 - is SAM a good representation/example? SAM segments an ROI, but bone fractures are lines.
- pg2;S2.1 - Maybe good to add one sentence here to mention Anthropic API may not be HIPAA compliant, and additional anon steps were taken.
- pg2;S2.1 - what is the cost per subject / total cost using Sonnet?
- pg3;S2.1.2 - quotation mark facing wrong way for (3)
- pg4;S2.2.1 - quotation mark facing wrong way for "guardrail"
- pg6;S2.5.1 - typo 'extractiona'

**Detailed Comments:**

see above

**Justification Of The Final Rating:**

The authors have addressed all my questions and concerns regarding the clarity of the manuscript, the experiment setting is now clearer with comments/feedback from all reviewers. The authors have also added statistical testing to their experiments. I therefore raise my rating from borderline to weak accept.

**Justification Of The Preliminary Rating:**

I don’t see anything methodologically incorrect that would warrant rejecting this paper. However, I can’t recommend acceptance either, as the experimental details are unclear that I have to make assumptions.

**Questions To Address In The Rebuttal:**

- I don't think there's anything methodologically incorrect, just overall clarification of the experiments and how the results are presented in writing.
- If possible, significance testing would really help to solidify the proposed method.

**Special Issue:**

No

---

> ### Author Response · Authors · 2025-03-08
>
> Thank you for your detailed review. We address your concerns point by point below:
>
> **pg4;S2.3** - yes 2939 is the number of reports
> **pg4;S2.3** - Thank you for identifying this ambiguity. We include our answer below in addition to the amended text in the manuscript. To clarify - each radiological examination included one report and multiple images (e.g. anteroposterior, lateral views).  For manual-finetuning we selected a random subset of 300 images that were annotated by an expert radiologist. These 300 images formed the basis of all reported results. For pseudo-blind finetuning only - we restricted the data to include the views where: (1) YOLO detected exactly one fracture in the selected view, (2) the LLM extraction confirmed only one fracture.
> **Figure 2** - We added an explicit reference to Figure 2 in the text.
> **Table 1** - We explicitly referenced Table 1 in the revised manuscript. The total count of 1,083 represents the overall number of fractures found in the analyzed images. The discrepancy in the "Fracture Pattern" category (total = 502) arises because radiologists did not specify a fracture pattern in all reports. For these unclassified cases, our automated extraction process assigned an "N/A" label, which we now explicitly document in a dedicated "N/A" row in the revised table.
> **pg6;S2.5.2;P2** - You're right, this is redundant text. We removed it in the revised manuscript.
> **pg6;S2.5.2;P2** - Thank you for highlighting this inconsistency. To clarify, the three datasets were constructed as follows: (1) The expert-annotated dataset (300 images) was randomly selected from the overall data available. Each image and report were manually reviewed by an expert radiologist and fractures were labelled with bounding boxes. (2) The small pseudo-annotated dataset (300 images) was then extracted from remaining images. (3) The large pseudo-annotated dataset (5,700 images) was taken as the remainder of the data. As mentioned in the answer above - each radiological report would have multiple images (e.g., anteroposterior, lateral views), which explains why the total number images is much larger than number of reports (2939).
> **Figure 3 and 4** - We added proper references to these figures in the text. Precision/recall/F1 data - we added the numbers into a new table - Table 2.
> **Figure 4** - The classes for precision/recall/F1 are of class ‘fracture’.
>
> Additional Clarifications:
> 1. All models were originally tested using the same 70 expert-annotated test images from dataset (a) for fair comparison.
> 2. Thank you for this great suggestion to perform significance testing. We had used our expert annotated dataset of 300 images for significance testing. The data was split with 8 folds using a sliding window approach with 50% overlap to perform cross-fold validation over  8 test datasets  of size 70 each. We evaluated all methods (Graz-pretrained, Manual Finetune, Pseudo-Blind Finetune Small and Pseudo-Blind Finetune Large) on these 8 x 70 separate sets of data to get mAP scores for the bounding box predictions, and corresponding Precision, Recall and F1 scores. This allowed us to measure mean and standard deviation of the performance and perform statistical tests comparing each method to the performance metrics from the reference (Graz-pretrained) method and report p-values indicating significant difference when p<0.05 as shown in the newly added table - Table 2. It is important to note - by using 8 cross-fold validation strategy - we obtained 8 sets of training data - which were used for 8 x Manual Finetune runs. Note also that we did NOT have to re-finetune the pseudo-blind models as they were never finetuned on the expert labelled data to start with.
> 3. Regarding rejected images in the pipeline (Figure 2): To ensure high-confidence of pseudo-labels, we excluded images where YOLO’s fracture detections conflicted with LLM-extracted fracture description from the radiology reports as described in our methodology. We accepted only if both methods agreed on the presence of a single fracture in that view.
>
>
> Minor Issues:
> **Abstract** - We removed citation references from the abstract.
> **pg2;S1;P3** - We revised the text to clarify that SAM is used for region segmentation. The core argument remains: pseudo-labeling methods (e.g. YOLO+LLM in our work, SAM in other contexts) aim to reduce annotation burdens in X-ray analysis.
> **pg2;S2.1** - Added sentence about HIPAA compliance.
> **pg2;S2.1** - The Claude 3.5 Sonnet API costs averaged 0.04 USD per report analysis, totaling approximately 117 USD for processing our dataset.
> **pg3;S2.1.2 and pg4;S2.2.1** - We fixed the incorrectly facing quotation marks.
> **pg6;S2.5.1** - We corrected the 'extractiona' typo.

---

> > ### Comment · Reviewer_Wupq · 2025-03-10
> >
> > Thank you for the revised revision, it certainly has improved clarity and understanding of the experiments conducted. Based on the revision, this is my understanding, is it correct?
> >
> > 2939 clinical examinations (2939 reports; 6300 images) -- also 2939 patients?
> >
> > 1) 300 image-report pairs manually annotated by radiologist
> >     - this is the ground truth data (test)
> >     - manual finetune method in table 2 uses this through cross-validation?
> >     - split into 8 folds with 50% overlap, giving 8 test sets of size 70 each
> >     - 35 * 8 = 315, final fold 15 images short? [NIT]
> >
> > 2) 300 Pseudo-Blind Finetune Small (300 reports?)
> > 3) 5700 Pseudo-Blind Finetune Large (2339 reports?)
> >
> > (2) and (3) are used as train/val and then tested on (1)?
> >
> >
> > Additional Comments:
> >
> > pg5;S2.4.1 - "The reference dataset was then divided into three parts: 214 images for fine-tuning the reference method, 16 images for validation, and 70 images for testing various methods, including the proposed method and comparison methods." -- redundant text if using cross val?
> >
> > pg6;S2.5.2;p2 - "There were two principal types of dataset." -- this seems out of place, is it needed? Also, the sentences that follows on from it I think is better if placed in S2.4.1 Data.
> >
> >
> > NITS:
> > - Table 2 - "Reference" in Method column is confusing with "Reference" dataset name, I think it's better just to use "Graz-Pretrained".
> > - figures should be place in the order they are referenced to in text (Figure 2 referenced before Figure 1)
> > - pg7;S2.6;p1 - reviewer DGcZ picked out text may be coppied from rebuttal? "As mentioned in the answer above"
> > - pg7;S2.6;p3 - incorrect latex rendering of "p¡0.05"

---

> > > ### Author Response · Authors · 2025-03-10
> > >
> > > > Thank you once again. We include our responses with inline comments below.
> > >
> > > Thank you for the revised revision, it certainly has improved clarity and understanding of the experiments conducted. Based on the revision, this is my understanding, is it correct?
> > >
> > >
> > > 2939 clinical examinations (2939 reports; 6300 images) -- also 2939 patients?
> > > > Yes that’s precisely correct
> > >
> > > 1. 300 image-report pairs manually annotated by radiologist
> > > - This is the ground truth data (test)
> > > > Exactly.
> > > - manual finetune method in table 2 uses this through cross-validation?
> > > > Yes, that’s correct!
> > > - split into 8 folds with 50% overlap, giving 8 test sets of size 70 each
> > > > Precisely, yes.
> > > - 35 x 8 = 315, final fold 15 images short? [NIT]
> > > > Not quite, we have a sliding window that wraps around to the beginning. In other words - if the fold finishes 15 images short - the algorithm wraps around to index 1-15.
> > >
> > > 2. 300 Pseudo-Blind Finetune Small (300 reports?)
> > > > Correct!
> > >
> > > 5700 Pseudo-Blind Finetune Large (2339 reports?)
> > > > Yes
> > >
> > > (2) and (3) are used as train/val and then tested on (1)?
> > > > Yes. That’s  exactly it.
> > >
> > > Additional Comments:
> > >
> > > pg5;S2.4.1 - "The reference dataset was then divided into three parts: 214 images for fine-tuning the reference method, 16 images for validation, and 70 images for testing various methods, including the proposed method and comparison methods." -- redundant text if using cross val?
> > > > You are right it is redundant. We will remove it from the final camera ready version.
> > >
> > > pg6;S2.5.2;p2 - "There were two principal types of dataset." -- this seems out of place, is it needed? Also, the sentences that follows on from it I think is better if placed in S2.4.1 Data.
> > > > Agree. We will move to 2.4.1 in camera ready.
> > >
> > > NITS:
> > > - Table 2 - "Reference" in Method column is confusing with "Reference" dataset name, I think it's better just to use "Graz-Pretrained".
> > > > Agree. Will change.
> > > - figures should be placed in the order they are referenced to in text (Figure 2 referenced before Figure 1).
> > > > Thank you for noticing it. We will swap the order of Figure 2 and Figure 1.
> > > - pg7;S2.6;p1 - reviewer DGcZ picked out text may be coppied from rebuttal? "As mentioned in the answer above"
> > > > Will update this. Correct.
> > > - pg7;S2.6;p3 -  incorrect latex rendering of "p¡0.05"
> > > > Thank you for noticing! We will fix it in camera ready submission.

---

> > > > ### Comment · Reviewer_Wupq · 2025-03-11
> > > >
> > > > I would like to thank the authors for thoroughly addressing my questions and concerns, particularly in adding statistical testing. The paper is in much better shape, and therefore I raise my final rating.
> > > >
> > > > As suggested by Area Chair AdCs, given the dataset's complexity (images, reports, patients), along with cross-validation partitions and network training hyperparameters, including these details in a table (maybe also with demographic stats) in the appendix would be helpful (just a suggestion, not a factor in my final decision).

---

### Author Rebuttal · Authors · 2025-03-08

**Rebuttal:**

We include here the revised manuscript with highlighted edits that reflect the changes made in response to reviewers' comments. Direct point by point responses to each question from each reviewer are posted in the main comment sections. Thank you to all reviewers for their valuable time and insightful feedback that helped to considerably improve the clarity of this manuscript.

**Supporting Material:**

/attachment/ce7624b9a9aca4b44d6c92c22ad789c8edcb4f33.pdf

---

### Meta-Review · Area_Chair_AdCs · 2025-03-20

**Recommendation:** Accept (Poster)
**Confidence:** 5

**Metareview:**

Reviewers were worried about some unclarities in the experimental section in the first submission, but they were all resolved after the rebuttal. One reviewer mentions that more extensive analysis, including edge case analysis and demographic analysis, could be included in future work but is satisfied with the current state for the context of the MIDL conference. Reviewers provided two weak acceptance scores and one strong acceptance score. I am recommending acceptance.